# Maternal High-Fat Diet Modulates *Cnr1* Gene Expression in Male Rat Offspring

**DOI:** 10.3390/nu13082885

**Published:** 2021-08-22

**Authors:** Dawid Gawliński, Kinga Gawlińska, Irena Smaga

**Affiliations:** Maj Institute of Pharmacology, Polish Academy of Sciences, Smętna 12, 31-343 Kraków, Poland; gawlin@if-pan.krakow.pl (D.G.); kingaw@if-pan.krakow.pl (K.G.)

**Keywords:** endocannabinoid, high-fat diet, rat offspring, *Cnr1*, CB1

## Abstract

In recent years, strong evidence has emerged that exposure to a maternal high-fat diet (HFD) provokes changes in the structure, function, and development of the offspring’s brain and may induce several neurodevelopmental and psychiatric illnesses. The aims of this study were to evaluate the effects of a maternal HFD during pregnancy and lactation on depressive-like behavior and *Cnr1* gene expression (encoding the CB1 receptor) in brain structures of rat offspring and to investigate the epigenetic mechanism involved in this gene expression. We found that a maternal HFD during pregnancy and lactation induced a depressive-like phenotype at postnatal days (PNDs) 28 and 63. We found that a maternal HFD decreased the *Cnr1* mRNA levels in the prefrontal cortex with the increased levels of *miR-212-5p* and methylation of CpG islands at the *Cnr1* promoter and reduced the level of *Cnr1* gene expression in the dorsal striatum with an increased level of *miR-154-3p* in adolescent male offspring. A contrasting effect of a maternal HFD was observed in the hippocampus, where upregulation of *Cnr1* gene expression was accompanied by a decrease of *miR-154-3p* (at PNDs 28 and 63) and *miR-212-5p* (at PND 63) expression and methylation of CpG islands at the *Cnr1* promoter in male offspring. In summary, we showed that a maternal HFD during pregnancy and lactation triggered several epigenetic mechanisms in the brains of rat offspring, which may be related to long-lasting alterations in the next generation and produce behavioral changes in offspring, including a depressive-like phenotype.

## 1. Introduction

In recent years, strong evidence has emerged that exposure to a maternal high-fat diet (HFD) provokes changes in the structure, function, and development of the offspring’s brain [1,2]. A maternal HFD may induce changes in the physiology of the offspring in early life, including metabolic syndrome, increased body weight, reduced leptin levels, and insulin resistance [2,3]. Moreover, the growing evidence links maternal malnutrition with both risk for and severity of several neurodevelopmental and psychiatric illnesses, including autism spectrum disorders, attention-deficit/hyperactivity disorder [4], cognitive impairment [5,6], schizophrenia [7], and depression [8,9,10]. 

The behavioral changes observed in early childhood may be associated with morphological, molecular, and functional alterations in the brain after a maternal HFD. Moreover, preclinical studies have shown that an HFD transgenerationally predisposes offspring to obesity and metabolic syndrome until the third generation via an epigenetic inheritance [11]. Genes probably can retain the memory of the early-life metabolic stress via epigenetic changes that include posttranslational modifications of histone proteins, noncoding RNAs, and DNA methylation, including downstream functional changes, and may alter target gene transcription or alternative splicing in tissues [12]. In fact, after maternal separation in mice, an increased level of methylation of the promoter-associated CpG island of the cannabinoid receptor type 1 (the CB1 receptor) (*Cnr1* gene) and methyl CpG-binding protein 2 (MeCP2 gene), a transcriptional regulator that binds methylated DNA, was reported in the second-generation brain [13]. Importantly, the above epigenetic mechanisms may contribute to individual differences in predisposition to depression. In this regard, a maternal diet during gestation and lactation can lead to long-lasting alterations in the epigenome, which may act as information to offspring across multiple generations [12], and induce long-term neurobiological modifications affecting synaptic function and structural plasticity [14]. 

The first preclinical reports, which appeared few years ago, showing antidepressant-like actions of substances that change the activity of endocannabinoid (eCB) system have led to the formulation of a new hypothesis concerning the etiology of depression and investigation of novel brain targets for new antidepressant drugs [15]. CB1 receptors are the primary cannabinoid receptors of the eCB system in the brain. CB receptors are metabotropic receptors that activate the G_i_ proteins and inhibit adenylate cyclase with an intracellular reduction in cAMP concentrations [16]. At the same time, CB1 receptor activation modulates gene expression, cell division, differentiation, and apoptosis in cells. Previously, we observed altered CB1 receptor expression in several brain structures in rats following an olfactory bulbectomy [17], while other authors detected lower density of CB1 receptors in a rat model of depression induced by stress [18,19,20,21]. Moreover, genetic deletion of CB1 receptors in mice results in a phenotype that strikingly resembles the profile of severe, typical depression; a similar depression-like behavioral phenotype was found after CB1 receptor blockade [22,23,24,25,26]. Our previous data also showed that pharmacological activation of the eCB system evokes antidepressant effects in rats [17,27,28,29]. 

Given the critical role of a maternal diet during pregnancy and lactation in producing a depression-like phenotype in offspring, we decided to perform induced genetic and epigenetic changes within the eCB system in rat brain structures. We analyzed *Cnr1* gene expression (which encodes the CB1 receptor) in adolescent (at postnatal day (PND) 28) and young adult (at PND 63) offspring whose mothers were fed on an HFD or a standard diet (SD) during pregnancy and lactation. We also evaluated the epigenetic mechanism by detecting the microRNAs (miRNAs), *miR-154-3p* and *miR-212-5p*, which are small noncoding RNAs that target the *Cnr1* gene to induce RNA silencing and post-transcriptionally regulate the expression of this gene. miRNAs were selected according to the base miRDB, which is an online database for miRNA target prediction and functional annotations, in which *miR-154-3p* (sequence AAUCAUACACGGUUGACCUAUU) possess a target score of 93%, while *miR-212-5p* (sequence ACCUUGGCUCUAGACUGCUUACUG) has a target score of 91% [30]. Finally, as gene methylation is thought to be one of the epigenetic mechanisms involved in the regulation of gene expression, we evaluated methylation of CpG islands at the *Cnr1* promoter in the structures with altered mRNA levels in offspring following a maternal HFD during pregnancy and lactation. We examined several brain structures that are either implicated in the pathogenesis of depression (i.e., prefrontal cortex (PFCTX) and hippocampus (HIP)) [31] or linked to anhedonia (i.e., dorsal striatum (DSTR)) [32] and represent sites affected by biochemical and morphological changes in depressed patients [31]. 

## 2. Materials and Methods 

### 2.1. Animals and Diets 

Wistar rats (Charles River, Sulzfeld, Germany) were housed in standard plastic cages in an animal colony room maintained at 22 ± 2 °C and 55 ± 10% humidity under a 12 h light-dark cycle (lights on at 6.00 a.m.). Animals had free access to food and water. Nulliparous female rats (200–240 g), after the acclimatization period and during the proestrus phase were mated with males. The gestation was confirmed by examining vaginal smears for the presence of sperm. Dams were individually housed and randomly assigned to two groups: SD (13% fat, 3.4 kcal/g; VRF1; Special Diets Services, Witham, UK) or HFD (60% fat, 5.31 kcal/g; C1057 mod.; Altromin, Lage, Germany). They all had free access to the diets during pregnancy (21 days) and lactation (21 days). The maternal HFD used in this study did not affect the litter size or birth weight of offspring [33]. After weaning, offspring at PND 22 were separated according to sex, housed 5 per cage, and switched to an SD. Male and female offspring were used in the present study (N = 8 rats/group). The present study was carried out in accordance with the European Union Directive 2010/63/EU and with the approval from the Local Ethics Commission at the Maj Institute of Pharmacology, Polish Academy of Sciences, Kraków, Poland (1270/2015; 17 December 2015). 

### 2.2. The Forced Swimming Test

The FST was conducted according to the protocol described previously [3,34]. Briefly, on the first day the 15 min pretest was performed. The rats were placed in a cylinder (height 50 cm × diameter 23 cm) filled with water to a depth of 30 cm (at temperature of 25 ± 1 °C). Then, the rats were removed from the cylinder, dried, and returned to their home cages. The cylinders were cleaned before subjecting each successive rat to the procedure. Twenty-four hours following the pretest, rats were tested for 5 min (300 s) under identical conditions. The test sessions were scored by two observers, and the immobility time was measured manually. A rat was rated to be immobile when it was making only movements necessary to keep its head above the water. 

### 2.3. Brain Structure Isolation 

Rat brain structures (PFCTX, DSTR, and HIP) were isolated according to the Rat Brain Atlas [35] at PND 28 or PND 63.

### 2.4. Molecular Analyses

#### 2.4.1. RNA/DNA Isolation

RNA and DNA were extracted using the AllPrep DNA/RNA/Protein Mini Kit (Qiagen, Hilden, Germany), following the manufacturer’s protocol. Briefly, the frozen tissues were homogenized (Bioprep-24 Homogenizer; Aosheng, Hangzhou, China). The homogenates were transferred to an AllPrep DNA spin column and centrifuged; then, the DNA spin column was used for DNA purification, while the flow-through was used for RNA purification. The total RNA and DNA concentrations were measured using an ND-1000 Spectrometer (NanoDrop Technologies Inc., Wilmington, DE, USA). 

#### 2.4.2. RT-qPCR Analyses

The synthesis of the cDNA by reverse transcription from equal amounts of RNA was performed using the High-Capacity cDNA Reverse Transcription Kit (Thermo Fisher Scientific, Life Technologies, Waltham, MA, USA). RT-qPCR was performed by using the Quant Studio 3 (Applied Biosystems, Foster City, CA, USA) and TaqMan Gene Expression Assays (Applied Biosystems, Waltham, MA, USA) for *Cnr1* (Rn00562880_m1). The PCR cycling conditions were as follows: an initial step at 95 °C for 10 min followed by 40 cycles at 95 °C for 15 s and then 60 °C for 60 s. The relative level of mRNA was assessed using the comparative CT method (2^−ΔΔCt^) and normalized to the level of the hypoxanthine phosphoribosyltransferase 1 (*HPRT1*), a housekeeping control (Rn01527840_m1). The values are expressed as the fold change relative to the control (n = 8 rats/group).

#### 2.4.3. Analyses of miRNA Expression

Total RNA (10 ng) and miRNA-specific stem-loop RT primers (rno-miR-154-3p, rno-miR-212-5p, Applied Biosystems, USA) were used for the reverse transcription. Then, the cDNAs were synthesized using the TaqMan MicroRNA Reverse Transcription Kit (Applied Biosystems, USA), according to the manufacturer’s instructions. RT-qPCR was conducted with the TaqMan MicroRNA assays, according to the protocol (Applied Biosystems, USA), to analyze the expression of the following miRNAs: *miR-154-3p* and *miR-212-5p*. The relative level of miRNA was assessed using the comparative CT method (2^−ΔΔCt^) and normalized to the level of the U6 small nuclear RNA (U6 snRNA) (n = 8 rats/group).

#### 2.4.4. Methylation of CpG Islands at the Cnr1 Promoter

Methylation-sensitive restriction qPCR analyses were performed using the EpiTect Methyl II PCR assays (Qiagen, Germantown, MD, USA), according to the manufacturer’s instruction. The method is based on the detection of the remaining input DNA after cleavage with methylation-sensitive and methylation-dependent restriction enzymes. Briefly, the restriction digestions for all samples were performed using a methylation-sensitive enzyme A (Ms), methylation-sensitive enzyme B (Md), both enzymes (Msd), or no enzymes (Mo) at 37 °C overnight. The reactions were stopped by heat-inactivation of the enzymes at 65 °C for 20 min. Next, the individual reactions for each of four digestions (Mo, Ms, Md, Msd) were prepared with the EpiTect Methyl II PCR Primer Assay for Rat Cnr1 (CpG Island 106518; EPRN106518-1A, Qiagen, USA) and RT2 SYBR Green ROX master mix on 96-well plates and performed using the Quant Studio 3 (Applied Biosystems, Foster City, CA, USA). Percentages of unmethylated and hypermethylated CpG values were calculated using a quantitation algorithm provided by the manufacturer (EpiTect Methyl II PCR Assay Handbook, Qiagen). 

### 2.5. Statistical Analyses

Statistical analyses were performed using the Student’s *t*-test. The GraphPad Prism 8 (GraphPad, La Jolla, CA, USA) was used to perform analyses and the data are presented as the mean ± SEM with individual value plots. *p* < 0.05 was considered statistically significant.

## 3. Results 

### 3.1. Behavioral Analyses

#### The Forced Swimming Test

Given the critical role of a maternal diet during pregnancy and lactation in depression, we used a rodent behavioral test for evaluating a depression-like phenotype in offspring. A maternal HFD during pregnancy and lactation increased the immobility time in both male and female adolescent and young adult offspring (Table 1). 

### 3.2. Molecular Analyses

#### 3.2.1. RT-qPCR Analyses

In search for the molecular changes in the brain that could be responsible for the risk of depressive-like behavior, we performed an analysis of *Cnr1* gene expression (which encodes the CB1 receptor) in the PFCTX, DSTR, and HIP of offspring whose mothers were fed on an HFD or SD during pregnancy and lactation. A maternal HFD during pregnancy and lactation decreased *Cnr1* gene expression in the PFCTX of adolescent male offspring (t = 2.891; df = 14; *p* = 0.0118) (Figure 1A), but did not change the mRNA level in female offspring at PND 28 (t = 0.9003; df = 14; *p* = 0.3832) (Figure 1B). The *Cnr1* mRNA levels were not affected in this structure in both male (t = 1.618; df = 14; *p* = 0.1279) (Figure 1C) and female (t = 0.121; df = 14; *p* = 0.9054) (Figure 1D) young adult offspring after a maternal HFD during pregnancy and lactation.

Similarly, a reduction in the *Cnr1* mRNA level was observed in the DSTR of adolescent male rats whose mothers were fed a HFD during pregnancy and lactation (t = 2.78; df = 14; *p* = 0.0147) (Figure 2A), but it was not changed in adolescent female offspring (t = 0.4335; df = 14; *p* = 0.6713) (Figure 2B) nor adult male (t = 0.3259; df = 14; *p* = 0.7494) (Figure 2C) and female (t = 1.766; df = 14; *p* = 0.0992) (Figure 2D) rats.

A contrasting effect of a maternal diet was observed in the HIP, where the *Cnr1* mRNA level was increased in male offspring at PND 28 (t = 2.673; df = 14; *p* = 0.0182) (Figure 3A) and PND 63 (t = 2.807; df = 14; *p* = 0.014) (Figure 3C). A maternal HFD did not change the *Cnr1* mRNA levels in adolescent (t = 1.02; df = 14; *p* = 0.3252) (Figure 3B) and adult (t = 1.205; df = 14; *p* = 0.2481) (Figure 3D) female offspring in this brain structure.

#### 3.2.2. miRNA Analyses

We also evaluated the maternal diet-induced epigenetic mechanism by detecting the miRNAs, that is *miR-154-3p* and *miR-212-5p*, which are small noncoding RNAs that target the *Cnr1* gene to induce RNA silencing and post-transcriptionally regulate the expression of this gene. A maternal HFD during pregnancy and lactation evoked an increase in *miR-212-5p* expression (t = 2.311; df = 14; *p* = 0.0366) in the PFCTX of adolescent male offspring (Figure 4B), but it did not affect the level of *miR-154-3p* (t = 1.003; df = 14; *p* = 0.3328) (Figure 4B) in this brain structure compared to that of male offspring whose mothers were fed on an SD.

Increased levels of *miR-154-3p* (t = 2.803; df = 14; *p* = 0.0141) (Figure 4C) and *miR-212-5p* (t = 5.343; df = 14; *p* = 0.0001) expression (Figure 4D) were observed in the DSTR of male offspring following a maternal HFD.

It was shown that a maternal HFD during pregnancy and lactation evoked a decrease in *miR-154-3p* (t = 18.76; df = 14; *p* < 0.0001) (Figure 4E) and *miR-212-5p* (t = 5.212; df = 14; *p* = 0.0001) (Figure 4F) expression in the HIP of adolescent male offspring compared to those of rats whose mothers were fed on an SD. In adult rats following a maternal HFD a reduction in *miR-154-3p* expression (t = 4.546; df = 14; *p* = 0.0005) was observed in the HIP, while the expression of the *miR-212-5p* (t = 1.212; df = 14; *p* = 0.2454) was not altered in this structure (Figure 4H).

#### 3.2.3. Methylation of CpG Islands at the Cnr1 Promoter

Finally, as it has been suggested that gene methylation represents one of the epigenetic mechanisms involved in the regulation of gene expression, we evaluated methylation of CpG islands at the *Cnr1* promoter in the structures with altered mRNA levels in offspring following a maternal HFD during pregnancy and lactation. Increased methylation of CpG islands at the *Cnr1* promoter was detected in the PFCTX (t = 4.554; df = 14; *p* = 0.0005) of adolescent offspring following a maternal HFD during pregnancy and lactation compared to that of offspring following a maternal SD (Figure 5A). At the same time, a maternal HFD did not change methylation of CpG islands at the *Cnr1* promoter in the DSTR (t = 0.414; df = 14; *p* = 0.6852) of rat offspring at PND 28 (Figure 5B).

Decreased methylation of CpG islands at the *Cnr1* promoter was observed in the HIP of both adolescent (t = 2.359; df = 14; *p* = 0.0334) (Figure 5C) and adult (t = 2.552; df = 14; *p* = 0.023) (Figure 5D) rats whose mothers were fed on an HFD during pregnancy and lactation compared to those whose mothers were fed a control diet.

## 4. Discussion

In the present study, we confirmed our previous observations [3,36] that a maternal HFD during pregnancy and lactation induces a depressive-like phenotype characterized by an increased immobility time in adolescent and young adult rats. These data are supported by other studies that revealed a depression-like phenotype in offspring linked to a maternal HFD during the lactation period [37] and pregnancy, lactation, and 14 weeks post-weaning [9]. From the molecular point of view, a maternal HFD decreased the *Cnr1* mRNA level in the PFCTX and DSTR of adolescent male offspring. A reduction in *Cnr1* gene expression was associated with an elevated level of *miR-212-5p* and an increased level of methylation of CpG islands at the *Cnr1* promoter in the PFCTX of rats at PND 28, while a decrease in the *Cnr1* level in the DSTR was related with the increased level of *miR-154-3p* in these rats. A contrasting effect of a maternal HFD was observed in the HIP, where upregulation of *Cnr1* gene expression was accompanied by a decrease of *miR-154-3p* and *miR-212-5p* expression and methylation of CpG islands at the *Cnr1* promoter in male adolescent rats. At the same time, in young adult rats, an increase in the *Cnr1* mRNA level was related with a decrease of *miR-154-3p* expression and methylation of CpG islands at the *Cnr1* promoter in the HIP after a maternal HFD during pregnancy and lactation.

Although a maternal HFD programs a similar depression-like phenotype in adolescent and adult offspring of both sexes, here we showed that molecular mechanisms involving the *Cnr1* gene and epigenetic markers were altered in a different sex-specific manner. There is evidence that biological sex is a predictive factor for lifetime prevalence of depression in women compared with men [38]: Women with depression are more likely to experience higher symptom severity and both sexes show differences in the presentation of biomarkers [39]. The behavioral differences between male and female offspring may be the result of complex neurological and synaptic alterations induced by a maternal HFD during pregnancy and lactation (for review please see [2]). Moreover, females appear to be more vulnerable than males to the forced swim test, and stress exposure results in decreases in serotonergic activity only in female rats [40], which may explain the gender differences observed on the molecular level within the eCB system.

The development and progression of a depression-like phenotype in offspring may be a result of the maternal diet-induced accumulation of genetic and epigenetic alterations. The PFCTX is a structure involved in emotional and social behavior and it controls a series of cognitive processes, all having as a final output the ability to adapt to various conditions. Additionally, the PFCTX plays a key role in many neuropsychiatric and neurodevelopmental disorders, which contribute to cognitive disabilities and emotional disturbances, including depression [29]. The principal development events (including a maternal diet during gestation and lactation) may affect the PFCTX maturation, lead to abnormal function, and impact behavioral changes in the offspring’s lifetime. In fact, in the present study, we showed that a maternal HFD during pregnancy and lactation reduced mRNA level of *Cnr1* in the PFCTX in male but not female adolescent rats compared to that of rats whose mothers were fed on a control diet. Similarly, male animals born from mothers fed on a palatable diet, and who continued with this diet after weaning, exhibited anxiety-like behavior and reduced expression of the eCB system, the main inhibitory retrograde input to glutamate synapses, reflected in a decrease of the *Cnr1* in the PFCTX of the offspring at adult age [41]. In preclinical studies, depressive-like behavior seems to be associated with impaired eCB signalization. Actually, a genetic deletion of CB1 receptors in mice results in a phenotype that strikingly resembles the profile of severe, typical depression; a similar depression-like behavioral phenotype was found after CB1 receptor blockade [22,23,24,25,26]. A reduced level of *Cnr1* gene expression was associated with the epigenetic changes in chromosomes that do not modify the sequence of DNA but may still lead to alterations in *Cnr1* gene expression. In fact, an increased level of *miR-212-5p* was observed in this structure, without alteration in the *miR-154-3p* levels, in male adolescent rats after a maternal HFD during gestation and lactation. It should be emphasized that *miR-154-3p* also affects a gene related to the insulin signaling pathway and mitogen-activated protein kinase signaling [42]. Another important epigenetic mechanism for the downregulation (silencing) of the expression of the gene *Cnr1* in the PFCTX involves DNA methylation at the promoter region. DNA methylation appears at the 5′ position of a cytosine residue followed by a guanine residue (CpG dinucleotide), which are often clustered (CpG islands) in the promoter regions of genes [43]. The presence of multiple methylated CpG sites in CpG islands of promoters causes stable silencing of genes. So, increased methylation of CpG islands at the *Cnr1* promoter induced by a maternal HFD during pregnancy and lactation may downregulate *Cnr1* gene expression in the PFCTX. In line with the present study, it was established that global DNA methylation in the PFCTX was altered following a perinatal HFD in male but not female offspring [44], which points to sex differences in changes in DNA methylation detected in offspring in response to a modified maternal diet. Contrastingly, a human study showed that the *Cnr1* level and DNA methylation at the *Cnr1* promoter did not change the peripheral blood mononuclear cells of patients with bipolar disorder (type I and II) and major depressive disorder; however, it should be noted that these subjects received long-term pharmacological treatment that may alter that gene expression [45].

Similarly, a maternal HFD during pregnancy and lactation evoked a decrease in the *Cnr1* mRNA level in the DSTR of male but not female adolescent rats. Our findings indicate that there are sex differences in the way a maternal and early life diet interact to alter brain *Cnr1* gene expression. The above data are in line with the evidence, suggesting that epigenetic regulation in the brain is often sex-specific and depends on estrogen receptor expression, neuroinflammatory signals, neonatal hormone exposure, and cellular differences in genetic sex [46]. A reduced *Cnr1* mRNA level in the DSTR seems to be associated with a weaker hedonic response (anhedonia), a symptom characteristic of a depressive state [20]. Several studies have shown that attenuation of the striatal eCB signaling provoked depressive-like behavior in stress-related animal models of depression [17,20,47,48,49] and in bulbectomized rats [17,50]. Moreover, our team demonstrated a reduced level of CB1 receptors in this structure in rats following olfactory bulbs removal [17]. Interestingly, we did not observe the alteration in DNA methylation at the *Cnr1* promoter in this structure after a maternal HFD; however, several types of epigenetic changes induced by a maternal HFD during pregnancy and lactation should be also considered, including DNA hypermethylation, loss of imprinting, and altered histone modification patterns, which contribute to altered gene expression in the DSTR. Silencing of a gene may also be initiated by small noncoding RNAs that target the *Cnr1* gene to induce RNA silencing and posttranscriptionally regulate the expression of this gene. In fact, a maternal HFD during pregnancy and lactation increased the levels of *miR-154-3p* and *miR-212-5p* in the DSTR of male rats at PND 28. The *miR-212-5p* overexpression decreased cell apoptosis, inflammation, and cytotoxicity [51] and prevented the death of dopaminergic neurons in a mouse model of Parkinson’s disease [52]. Crosstalk between the eCB and dopaminergic system in depression is well known. It has been revealed that altered dopaminergic neurotransmission provokes anhedonia [53], while endogenous dopamine is also involved in the generation of eCB-long term depression (LTD) by synaptic activity via striatal D2 receptors [54]. Additionally, following the olfactory bulbectomy an increase in dopamine release and D1 and D2 receptor genes expression were observed in the striatum [53,55], while alterations of this neurotransmitter evoke a decrease in the endocannabinoid release via D1 receptors [56]. It should be noted that the reduced *Cnr1* gene expression level in the PFCTX and DSTR did not persist until adulthood. Probably efficient compensatory mechanisms normalize the level of the *Cnr1* in these structures, while the depression-like phenotype was also displayed by young adult rats. Similarly, it was shown that a maternal Western diet (rich in fat) during gestation and lactation reduced the level of this gene expression in the hypothalamus, which was observed only at PND 25, and the normalization of the *Cnr1* mRNA level occurred in older rats [57]. It seems that the reduced *Cnr1* level may be one of the mechanisms involved in inducing a depression-like phenotype in adolescent male rats, but further study will be needed to explain the long-lasting alterations in offspring triggered by a maternal diet.

Interestingly, a maternal HFD during pregnancy and lactation increased the *Cnr1* mRNA level in the HIP in adolescent male offspring rats, and this change persisted even to adulthood despite the lack of exposure of the offspring to an HFD after the lactation period (change to an SD after weaning). The latter change correlates well with the higher levels of CB1 receptors in male offspring at birth [58], and such upregulation of CB1 receptors was observed in the hypothalamus and was induced by maternal exposure to an HFD during the eight weeks before mating and throughout gestation and lactation. Another study showed that a maternal HFD during the eight weeks before mating and throughout gestation increased the *Cnr1* level in the hypothalamus of both sexes—male and female—at birth with leptin pathway impairment, which might contribute to increased levels of *Cnr1* mRNA [59]. Additionally, in a genetic-induced model of depression, the increased CB1 receptor level was related to depressive-like behavior in Wistar Kyoto rats [17,60]. In contrast, a maternal HFD enriched in omega-3 fatty acids during the last week of gestation and through lactation reduced levels of 2-arachidonoylglycerol (2-AG) in the HIP and arachidonic acid (product of the eCBs degradation) in the HIP and hypothalamus in ten-day-old pups [61]. Omega-3 fatty acids play a crucial role during neurodevelopment, and they are involved in transmembrane receptor function, gene expression, neuroinflammation, as well as neuronal differentiation and growth [62], while deficiency of these fatty acids in mothers during gestation has been associated with maternal depression and childhood neurodevelopmental disorders [63]. On the other hand, based on the fact that CB1 receptors regulate food intake, the increased level of *Cnr1* gene expression in the HIP of male offspring may be associated with altered body mass in the rats following a maternal HFD during pregnancy and lactation. In fact, we recently presented that exposure to a maternal HFD during gestation and lactation provoked body mass increase in male offspring at PND 63 compared to the weight of the rats in the SD group [3]. It should be mentioned that the latter change was not observed in female offspring [3], which correlates well with no alterations in the hippocampal *Cnr1* level in female offspring following a maternal HFD in the present study.

Consistent with the upregulation of the *Cnr1* gene reported in the present study, we observed a significant reduction of DNA methylation at the *Cnr1* promoter in the HIP of both adolescent and young adult rats. It was documented that a maternal HFD during the eight weeks before mating and throughout gestation increased the histone acetylation percentage of the *Cnr1* promoter in male offspring and increased the androgen receptor binding to the *Cnr1* promoter, which can contribute to higher expression of *Cnr1* in newborn offspring [59]. Upregulation of the *Cnr1* gene in the HIP seems to also be related to the reduced *miR-154-3p* level in male offspring at PND 28 and PND 63, while a maternal diet-induced reduction in the *miR-212-5p* level was observed only in adolescent offspring. Of note, one miRNA can regulate multiple mRNAs [64], suggesting that dysregulation of a single miRNA can have a dramatic downstream effect. In fact, *miR-212-5p* is highly expressed in the brain and is especially important for neuronal function, synaptic plasticity, memory formation, and blood–brain barrier integrity [65]. It is worth mentioning that *miR-212-5p* was identified as a target of the cAMP response element-binding protein (CREB) and REST, a regulator of neuronal transcription that controls its own expression [66]. The regulation of CREB-dependent transcription may contribute to constraining long-term plasticity, while *miR-212/132* knockout mice have impaired memory [67].

In summary, we showed that a maternal HFD during pregnancy and lactation induces sex-related changes in the offspring, disrupting normal expression of the gene encoding the CB1 receptor in the brain structures important for the pathogenesis of depression. The altered *Cnr1* gene expression in adolescent male offspring may result from a disturbed maternal HFD epigenetic pattern in these animals (altered promoter methylation of this gene and expression of selected miRNAs) and could be a potential explanation for the observed depressive-like behaviors. At the same time, further research is needed to elucidate the molecular mechanism responsible for the development of depressive-like behavior in female offspring exposed to a maternal HFD that do not display changes in the expression of the *Cnr1* gene. It seems very important to study both males and females in the context of searching for contributory factors in the development of mental brain disorders.

## Figures and Tables

**Figure 1 nutrients-13-02885-f001:**
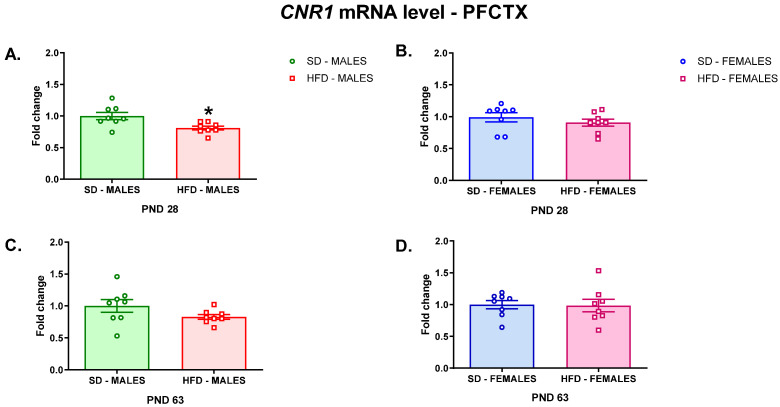
Changes in the *Cnr1* mRNA level in the PFCTX of adolescent (**A**) male and (**B**) female offspring or young adult (**C**) male and (**D**) female offspring whose mothers were fed on an SD or HFD during pregnancy and lactation. HFD—high-fat diet; PFCTX—prefrontal cortex; PND—postnatal day; SD—standard diet. Data are presented as the mean ± SEM with individual value plots. N = 8 rats/group. * *p* < 0.05 vs. SD.

**Figure 2 nutrients-13-02885-f002:**
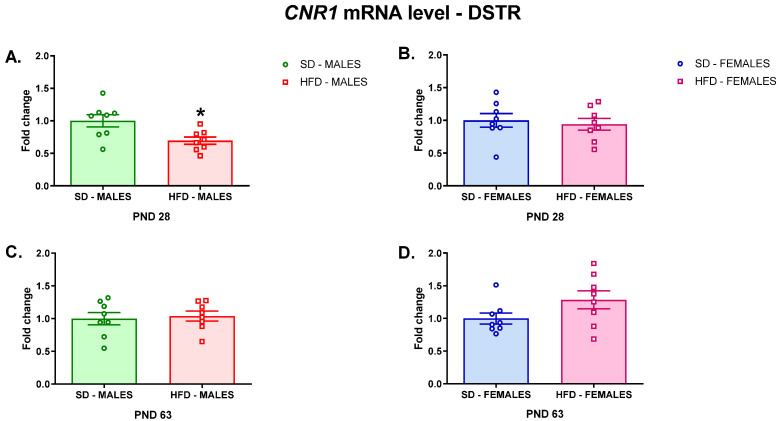
Changes in the *Cnr1* mRNA level in the DSTR of adolescent (**A**) male and (**B**) female offspring or young adult (**C**) male and (**D**) female offspring whose mothers were fed on an SD or HFD during pregnancy and lactation. DSTR—dorsal striatum; HFD—high-fat diet; PND—postnatal day; SD—standard diet. Data are presented as the mean ± SEM with individual value plots. N = 8 rats/group. * *p* < 0.05 vs. SD.

**Figure 3 nutrients-13-02885-f003:**
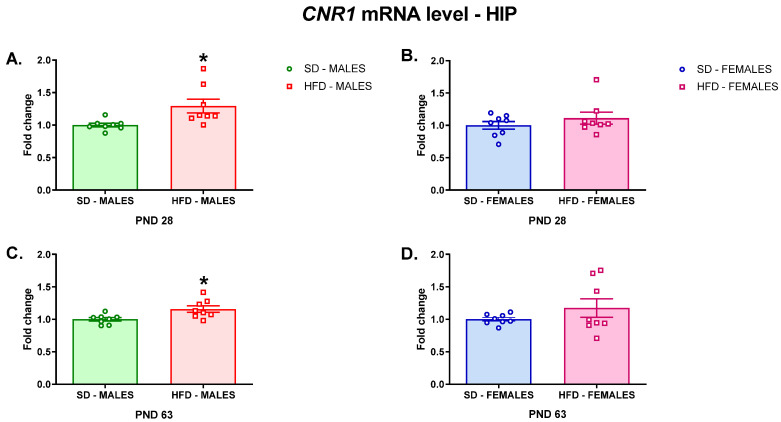
Changes in the *Cnr1* mRNA level in the HIP of adolescent (**A**) male and (**B**) female offspring or young adult (**C**) male and (**D**) female offspring whose mothers were fed on an SD or HFD during pregnancy and lactation. HFD—high-fat diet; HIP—hippocampus; PND—postnatal day; SD—standard diet. Data are presented as the mean ± SEM with individual value plots. N = 8 rats/group. * *p* < 0.05 vs. SD.

**Figure 4 nutrients-13-02885-f004:**
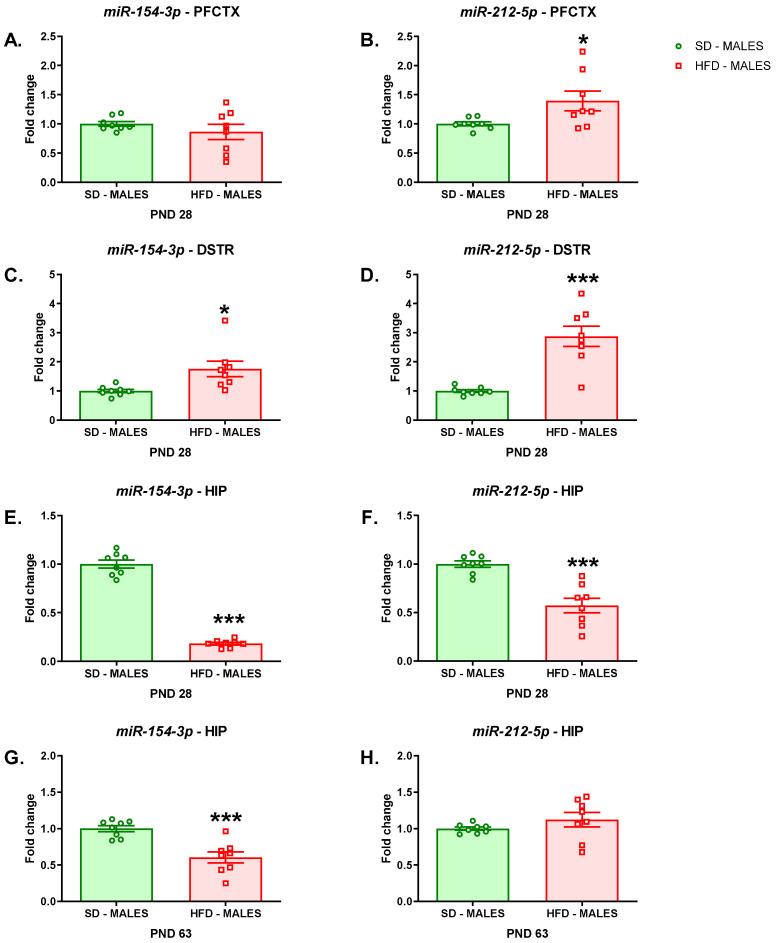
Effects of a maternal HFD during pregnancy and lactation on the levels of *miR-154-3p* and *miR-212-5p* in the (**A**), (**B**) PFCTX, (**C**), (**D**) DSTR, and (**E**), (**F**) HIP of adolescent offspring and in the (**G**), (**H**) HIP of young adult offspring. DSTR—dorsal striatum; HFD—high-fat diet; HIP—hippocampus; PFCTX—prefrontal cortex; PND—postnatal day; SD—standard diet. Data are presented as the mean ± SEM with individual value plots. N = 8 rats/group. * *p* < 0.05; *** *p* < 0.001 vs. SD.

**Figure 5 nutrients-13-02885-f005:**
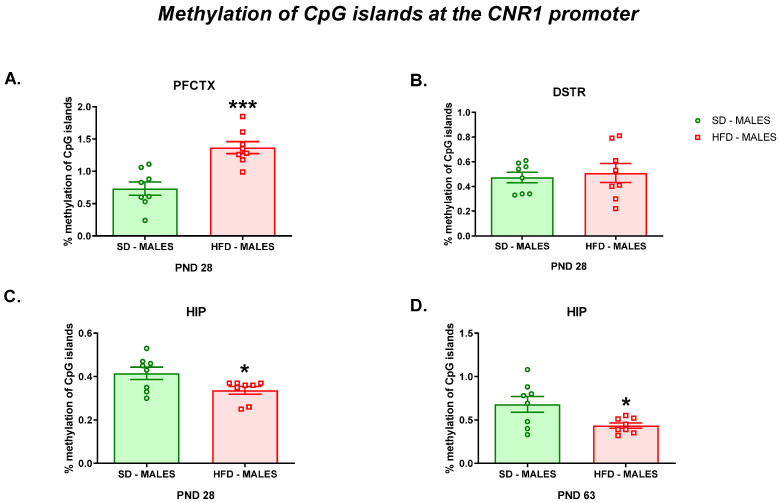
Effects of a maternal HFD during pregnancy and lactation on the methylation of CpG islands at the *Cnr1* promoter in the (**A**) PFCTX, (**B**) DSTR, and (**C**) HIP of adolescent offspring and in the (**D**) HIP of young adult offspring. DSTR—dorsal striatum; HFD—high-fat diet; HIP—hippocampus; PFCTX—prefrontal cortex; PND—postnatal day; SD—standard diet. Data are presented as the mean ± SEM with individual value plots. N = 8 rats/group. * *p* < 0.05; *** *p* < 0.001 vs. SD.

**Table 1 nutrients-13-02885-t001:** The immobility time in the forced swimming test in male and female offspring rats whose mothers were fed on an HFD at postnatal days (PNDs) 34 and 69.

Group	Sex	Immobility Time (s)		Statistical Analyses
		SD	HFD	
adolescent	male	173.5 ± 11.47	209.8 ± 2.88 **	t = 3.073; df = 14; *p* = 0.0083
	female	151.9 ± 11.15	197.0 ± 9.86 **	t = 3.033; df = 14; *p* = 0.0089
“young” adult	male	220.6 ± 5.22	246.5 ± 5.44 **	t = 3.440; df = 14; *p* = 0.004
	female	226.7 ± 9.23	260.5 ± 4.63 **	t = 3.277; df = 14; *p* = 0.0055

HFD—high-fat diet; SD—standard diet. All data are expressed as the mean ± SEM. N = 8 rats/group. ** *p* < 0.01 vs. SD.

## Data Availability

All data relevant to the study are included in the article.

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
