# Peer review of "Maternal High-Fat Diet Modulates Cnr1 Gene Expression in Male Rat Offspring"

_nutrients, 2021, doi:10.3390/nu13082885_

Round 1

Reviewer 1 Report

The authors present an original study focusing on the effects of maternal diet on the expression of the Cnr1 gene in specific brain areas of the offspring, distinguishing between males and females.

The search for the causes of the depressive behavior manifested by the offspring in molecular and epigenetic events concerning the Cnr1 gene represents the originality of the study.

The study was conducted with rigor and the results are well presented, however some weaknesses must be considered:

  • the observed changes in the expression of the Cnr1 gene, in all of the investigated brain areas, are small in entity, less than 50% (vs. Controls). May these variations have direct significant biological implications? No evidence is provided on the protein level of the CB1 receptor. No evidence of variations directly dependent on a CB1 receptor signaling deficiency is provided (neither at the molecular level, nor at the systemic level). Did the authors collect data on body weight and the feeding behavior of the studied offspring? Are there any differences between males and females? In the reviewer's opinion, the correlation between CB1 receptor mRNA levels and the observed depressive behavior is speculative.
  • the authors observed a marked sex-dependence of the alterations in the expression of the Cnr1 gene, however both male and female adolescent and young adult offspring showed increased immobility time during the forced swimming test. This is little or not at all discussed. Investigations to explain it are missing.
  • The depressive-like phenotype of the offspring of mothers fed a HFD during pregnancy and lactation could be the result of complex neurological and synaptic alterations; this aspect was not investigated and only poorly discussed.
  • the discussion is somewhat speculative. The conclusions are weak and not supported by the results.
  • editing of English language and style is required throughout the manuscript (for example, authors should check lines 11-12, line 14, 44, 71, 72, 162, lines 164-165, line 268, 299)

Minor comments:

  • Introduction section: not clear for the reader the reason why studying miR-154-3p and miR-212-5p. This should be better explained and supported on the basis of the scientific literature. Appropriate references should also be cited.
  • Materials and Methods section: RT-qPCR analyses, the mathematical algorithm used for the calculation of the relative mRNA expression is not reported. Authors should specify. Statistical analyses, English language and style of this paragraph need editing.

Author Response

Dear Reviewer,

Please find attached our revised manuscript “Maternal high-fat diet modulates Cnr1 gene expression in male rat offspring”. We would like to thank the Reviewer for the assessment of the manuscript as well as for the helpful comments. We have followed all the Reviewer’s requirements.

Responses to the Reviewer #1:

The study was conducted with rigor and the results are well presented, however some weaknesses must be considered:  the observed changes in the expression of the Cnr1 gene, in all of the investigated brain areas, are small in entity, less than 50% (vs. Controls). May these variations have direct significant biological implications? No evidence is provided on the protein level of the CB1 receptor. No evidence of variations directly dependent on a CB1 receptor signaling deficiency is provided (neither at the molecular level, nor at the systemic level).

Response:

We agree with the Reviewer comment about the expression of the Cnr1 gene in the investigated structure, however in our experience and based on the scientific reports changes in the gene expression at the level of less than 50% possess behavioral implications (Smaga et al, 2020; Gawlinska et al, 2020, Sales et al., 2021). The aim of this study was to present the maternal diet-induced genetic and epigenetic changes within the endocannabinoid system, while the variations directly dependent on a CB1 receptor signaling deficiency have not been evaluated in the present study, however, we will take this important issue in further investigation.

Did the authors collect data on body weight and the feeding behavior of the studied offspring? Are there any differences between males and females? In the reviewer's opinion, the correlation between CB1 receptor mRNA levels and the observed depressive behavior is speculative.

Response:

In the revised version of the Discussion we have addressed the Reviewer’s remark (the revised version: page: 11; lines: 391-399).

    the authors observed a marked sex-dependence of the alterations in the expression of the Cnr1 gene, however both male and female adolescent and young adult offspring showed increased immobility time during the forced swimming test. This is little or not at all discussed. Investigations to explain it are missing. The depressive-like phenotype of the offspring of mothers fed a HFD during pregnancy and lactation could be the result of complex neurological and synaptic alterations; this aspect was not investigated and only poorly discussed, the discussion is somewhat speculative. The conclusions are weak and not supported by the results.

Response:

According to the Reviewer suggestion, several comments regarding sex differences have been added to the Discussion, as well as the Summary section has been rewritten (the revised version: page: 8-9, lines: 280-291; page: 9, lines: 324-328; page: 10-11, lines: 334-339; page: 11, lines: 391-399; page: 11, lines: 417-428).

    editing of English language and style is required throughout the manuscript (for example, authors should check lines 11-12, line 14, 44, 71, 72, 162, lines 164-165, line 268, 299)

Response:

The whole manuscript has been corrected by a native English speaker.

Minor comments:

    Introduction section: not clear for the reader the reason why studying miR-154-3p and miR-212-5p. This should be better explained and supported on the basis of the scientific literature. Appropriate references should also be cited.

Response:

According to the Reviewer suggestion, the information about the miRNAs used in the present study has been added to the Introduction section (the revised version: page: 2, lines: 77-81).

    Materials and Methods section: RT-qPCR analyses, the mathematical algorithm used for the calculation of the relative mRNA expression is not reported. Authors should specify. Statistical analyses, English language and style of this paragraph need editing.

Response:

As requested, the mathematical algorithm used in calculation of the relative mRNA expression has been added (page: 3, lines: 135-137). Additionally, the whole manuscript has been corrected by a native English speaker.

We hope that the present version of our manuscript is suitable for publication in Nutrients. Once again, we would like to thank the Reviewer for all the suggestions which have allowed us to make constructive corrections.

Yours sincerely,

Irena Smaga, PhD

Reviewer 2 Report

In the manuscript Maternal high-fat diet modulates Cnr1 gene expression in male rat offspring, the authors describe phenotypic changes present in the offspring of mice fed high fat diets.  These changes are then linked to epigenetics changes observed in the offspring.  The authors have done a thorough job in the preparation of their data and the presentation of their figures, and the data itself seems very convincing.  However, the text itself would benefit from additional information and more effort to walk the reader through the experiments.  Additionally, there are some concerns about the conclusions being drawn from the data presented here.  It is this reviewer’s opinion that there are several issues (one major) that need to be addressed, mostly through editing and clarification, before the manuscript is ready for publication.  These specific concerns are outlined below:   

Major Concerns

The biggest issue I have with this paper is the problems created by the fact that you observe no epigenetic changes in the female rats.  This result seems to chip away at the idea that you are successfully linking phenotypic response to the observed epigenetic changes.  Specifically, if you see no change in the mRNA levels of HFD offspring females, why do they demonstrate a higher immobility time, which is your readout for depression?  How can immobility time and protein expression even be linked if the males show a change in mRNA levels, females do not, and yet they have the same phenotype?  To your credit, you’re very careful to state that your findings only apply to the male rats, but the discrepancy in your findings with the female rats needs to be discussed in more depth.

Moderate Concerns 

Overall, the explanation of your results in your Results section seems to be minimalistic.  Please briefly describe the experiment and its goals before presenting data, to help lead the reader through the thought process of these experiments.

The following statement in your introduction should have a citation(s) associated with it: “Genes probably can retain the memory of the early-life metabolic stress via epigenetic changes that include posttranslational modifications of histone proteins, noncoding RNAs, and DNA methylation, which downstream functional changes and may.”

Similarly, there should be a citation associated with: “Importantly, the above epigenetic mechanisms may contribute to individual differences in predisposition to depression.”

It would be useful to comment on why, potentially, the effect of HFD on CNR1 mRNA levels in the PFCTX and DSTR are lost at 63 days.

Tied to the major concern of this paper, it would be worthwhile to comment more on why there might be gender differences in your results.

Minor Concerns

Line 71: “We decided to perform epigenetic changes within the eCB system in rat brain structures.”  More accurately, you “induced epigenetic changes.” 

Author Response

Dear Reviewer,

Please find attached our revised manuscript “Maternal high-fat diet modulates Cnr1 gene expression in male rat offspring”. We would like to thank the Reviewer for the assessment of the manuscript as well as for the helpful comments. We have followed all the Reviewer’s requirements.

Responses to the Reviewer #2:

Major Concerns: The biggest issue I have with this paper is the problems created by the fact that you observe no epigenetic changes in the female rats.  This result seems to chip away at the idea that you are successfully linking phenotypic response to the observed epigenetic changes.  Specifically, if you see no change in the mRNA levels of HFD offspring females, why do they demonstrate a higher immobility time, which is your readout for depression?  How can immobility time and protein expression even be linked if the males show a change in mRNA levels, females do not, and yet they have the same phenotype?  To your credit, you’re very careful to state that your findings only apply to the male rats, but the discrepancy in your findings with the female rats needs to be discussed in more depth.

Response:

According to the Reviewer suggestion, several comments regarding sex differences have been added to the Discussion, as well as the Summary section has been rewritten (the revised version: page: 8-9, lines: 280-291; page: 9, lines: 324-328; page: 10-11, lines: 334-339; page: 11, lines: 391-399; page: 11, lines: 417-428).

Moderate Concerns: Overall, the explanation of your results in your Results section seems to be minimalistic.  Please briefly describe the experiment and its goals before presenting data, to help lead the reader through the thought process of these experiments.

Response:

According to the Reviewer suggestion, the explanation of the Results and its goals have been supplemented (the revised version: page: 4, lines: 170-172, page: 4-5, lines: 180-183; page: 6, lines: 219-222; page: 7, lines: 243-246).

The following statement in your introduction should have a citation(s) associated with it: “Genes probably can retain the memory of the early-life metabolic stress via epigenetic changes that include posttranslational modifications of histone proteins, noncoding RNAs, and DNA methylation, which downstream functional changes and may.”

Similarly, there should be a citation associated with: “Importantly, the above epigenetic mechanisms may contribute to individual differences in predisposition to depression.”

Response:

As suggested, we have added the references (the revised version: page: 2, lines: 44 and 52-53)

It would be useful to comment on why, potentially, the effect of HFD on CNR1 mRNA levels in the PFCTX and DSTR are lost at 63 days.

Response:

In the revised version of the Discussion we have addressed the Reviewer’s remark (the revised version: page: 11; lines: 362-371):

It should be noted that the reduced Cnr1 gene expression level in the PFCTX and DSTR did not persist until adulthood. Probably efficient compensatory mechanisms normalize the level of the Cnr1 in these structures, while the depression-like phenotype was displayed also by young adult rats. Similarly, it was shown that a maternal western diet (rich in fat) during gestation and lactation reduced the level of this gene expression in the hypothalamus, that was observed only on PND 25, and the normalization of the Cnr1 mRNA level occurred in older rats [57]. It seems that the reduced Cnr1 level may be one of the mechanisms involved in inducing a depression-like phenotype in adolescent male rats, but further study will be needed for the explanation of the long-lasting alterations triggered by a maternal diet in offspring.

Tied to the major concern of this paper, it would be worthwhile to comment more on why there might be gender differences in your results.

Response:

According to the Reviewer suggestion, several comments regarding sex differences have been added to the Discussion (the revised version: page: 8-9, lines: 280-291; page: 9, lines: 324-328; page: 10-11, lines: 334-339; page: 11, lines: 391-399).

Minor Concerns Line 71: “We decided to perform epigenetic changes within the eCB system in rat brain structures.”  More accurately, you “induced epigenetic changes.”

Response:

We have corrected the indicated phrase into more precise one (the revised version: page: 2, line: 70).

We hope that the present version of our manuscript is suitable for publication in Nutrients. Once again, we would like to thank the Reviewer for all the suggestions which have allowed us to make constructive corrections.

Yours sincerely,

Irena Smaga, PhD

Round 2

Reviewer 2 Report

The authors have adequately addressed my initial concerns about the manuscript.  Thank you for your work to clarify these points.